# Radiation-Induced Patterning at the Nanometric Scale: A Phase Field Approach

**DOI:** 10.3390/ma15092991

**Published:** 2022-04-20

**Authors:** David Simeone, Philippe Garcia, Laurence Luneville

**Affiliations:** 1Université Paris Saclay, CEA, Service de Recherche en Metallurgie Appliquée, F-91191 Gif sur Yvette, France; 2CEA, DES, IRESNE, DEC, F-13108 Saint Paul Lez Durance, France; philippe.garcia@cea.fr; 3Université Paris Saclay, CEA, Service d’Etude des Réacteurs et de Mathematiques Appliquées, F-91191 Gif sur Yvette, France; laurence.luneville@cea.fr

**Keywords:** radiation-induced patterning, phase field approach, nano-structures

## Abstract

The phase field approach was developed in the last 20 years to handle radiation damage in materials. This approach bridges the gap between atomistic simulations extensively used to model first step of radiation damage at short time and continuum approach at large time. The main advantage of such an approach lies in its ability to compute not only the microstructure at the nanometric scale but also to calculate generalized susceptibilities such as elastic constants under irradiation. After a brief description of the rate theory, used to model the microstructure induced by irradiation, we briefly discuss the foundation of the phase field method, highlighting not only its advantages, but also its limitations in comparison with the rate theory. We conclude this presentation by proposing future orientations for computing the microstructure in irradiated materials.

## 1. Introduction

Modern materials science has enormously progressed from a descriptive endeavor to more physically sounded foundations. The ultimate goal of such a fundamental tilt is to design material with tunable properties. The use of new experimental techniques such as particle irradiation made possible to escape the tyranny of the phase diagram. Overcoming this restriction, it becomes possible to observe the formation of patterns at the nanometric scale. It is now clear that such patterns affect the properties of materials and thus need to be understood and controlled.

On the other hand, materials under irradiation may be considered as toy models to study systems maintained far from equilibrium. Varying the control parameters (temperature, particles flux), experimentalists can then test different approaches for modeling instabilities and pattern formation far from equilibrium.

The slowing down of energetic particles (electrons, ions and neutron) in crystalline solids were studied in detail over the past 60 years [1]. Even if mechanisms occurring during the slowing down of particles are complex [2], it is nowadays admitted that a part of the energy dissipated by incident particles during their slowing down generates a distribution of point and extended defects in materials. This distribution is far from being uniformly distributed along the material but localized in spatial areas, named displacement cascades [3]. The atomic fraction of defects in these areas is largely superior to those expected from the thermal equilibrium. Such an over-concentration of defects is responsible for the formation of unexpected microstructures over few tens of nanometers: voids, bubbles and nano-precipitates. Moreover, self-organized nano-structures may emerge under some specific conditions of irradiation flux and temperature. Many examples of such patterns have been observed in metals, alloys, ceramics and amorphous materials [3,4,5,6,7].

The conditions associated with the formation of these radiation-induced nano-patterns, as well as mechanisms associated with their formation is still subject of debate. In this paper, we present new achievements to answer these questions applying the Phase Field formalism. The first part of this work is devoted to the rate theory approach and its extension applied to discuss the formation of nano-patterns. A brief description of this technique, its successes and limitations is discussed. The second part of this work describes in detail the foundation of the Phase Field method, its application to radiation damage and its limitation. Some prospects for future works is proposed as a conclusion.

## 2. The Rate Theory

The formation of steady state patterns in driven chemical systems has long been a puzzling problem, both theoretically and experimentally. In 1952, Turing [8] proposed that the interplay between diffusion and reacting species could lead to the formation of spatial patterns due to non linear interactions. The interest of this original approach is to describe the dynamics of the system under investigation by sets of ordinary or partial differential equations. Conditions and mechanisms associated with the formation of chemically driven systems can thus be associated with the stability of these dynamical equations. Patterns result from Hopf or Turing bifurcations [2,5]. Extension of this approach to radiation damage in metals was developed in the early 70 [9,10].

### 2.1. Foundation of the Rate Theory

The aim of the rate theory was first developed to study the radiation-induced microstructure at large time such as for instance the radiation-induced swelling of iron steel or nuclear fuels. Two major assumptions are applied to build up a rate theory:Defects produced during the slowing down of incident particles in the medium are produced over a time scale $\tau_{C}$, i.e., the characteristic time of a displacement cascade ($10^{-9}$ s), much smaller than the time scale $\tau_{D}$ associated to the growth or shrink of extended defects (voids, dislocations) which is controlled by mass transport ($10^{-6}$ s). Such an assumption makes a distinction between point defects (vacancy, interstitials) and extended defects (voids, dislocations, self interstitial tetrahedron ) produced in displacement cascades.The flux of incident particle is large enough to neglect the correlation between sub-cascades and the the correlation between produced defects. The production rate of point defects can then be considered as uniform in the irradiated area.

Under these two assumptions, a master equation may be formulated for modeling the dynamics of extended defects. Such a formulation of the problem can be obtained performing a coarse-graining over a space scale larger than the sub cascade size (few angstroms). The master equation holds over a characteristic time scale larger than the displacement time scale of a cascade formation [11]. By analogy with the Doring and Becker formalism of nucleation [5], the size of each extended defect, i.e., void cavities for instance, grows and shrinks by a process of accretion and emission, controlled by the fluctuation of point defects in its neighborhood [12]. Thus, an appropriate framework for describing such events is the theory of stochastic processes [13,14] for which the size *x* of extended defects appears to be a random variable defined by a probability P(x,t) in the continuum approximation:(1)P(x,t)=∫dx∫dtW(x+δx→x,t)P(x+δx,t)−W(x→x+δx,t)P(x,t)
where δx refers to the indexation of different states and W(x→x+δx,t) is the transition rate from state x+δx to state *x*. This description is extensively used to discuss a large class of phenomena such as chemical reactions [13] or in nucleation process under thermal equilibrium as pointed out by Becker and Doring (for a summary of the theory of nucleation under thermal equilibrium, see the book of Ghoniem [5]).

To make things more concrete, we can determine the transition rate produced under ballistic mixing. Applying the seminal approach of Sigmund [2,11], it reduces to:(2)$$\begin{array}{ccc} W(x+\delta x\to x,t) & = & p(x,\delta x,t)\\ & = & p(x,\delta x)\delta (t) \end{array}$$

Since the life time of a displacement cascade is of about $10^{-9}$ seconds [1], W(x+δx→x,t) can be considered as a instantaneous function of time, i.e., its time dependence reduces to a Dirac. Under irradiation, the characteristic relocation length of a recoil depend on its kinetic energy and then varies. The function p(x,δx,t) can thus be approximated by a probability density function $p_{R}\left(x\right)$ for high energy displacement cascades, i.e., displacement cascades produced for high energetic particles (E>10 keV.) The term *R* is a characteristic relocation distance which can be understood as the mean free path for the atomic relocation. Its second moment $\int x^{2}p_{R}\left(x\right)dx$ is the ballistic diffusion coefficient and can be computed from binary collision approximation [11] or extracted from Molecular Dynamics simulations.

It must be kept in mind that the kinetic energy of ion beams, used for simulating radiation damage produced in nuclear plants [6], are of about 10 keV/amu, well below the Coulombian barrier, and they do not induce any fission nor disintegration in the materials. In nuclear plants, neutrons will not only generate damage, but also they will modify the chemical composition of the material [15]. These chemical modifications are not taken into account in this paper.

### 2.2. The Rate Theory Approach

It has been clearly shown [13] that if the process W(x,t) modeling the effect of sub cascades on the microstructure of materials under consideration is a Poisson distribution, *i.e* describes a rare event, resolution of the Master Equation can be simplified. For a Poisson distribution, the absorption and emission rates depend no more on P(n,t) but only on <x> the first moment of the random variable *x*, i.e., the mean value of the distribution P(x,t). The dynamics associated to the growth of the cluster is no more given by a set of coupled differential equations but reduces to a simple differential equation describing the dynamics of the averaged cluster size <xP(x,t)>. In other words, it is possible to compute from the Master Equation, the first moment of the distribution according to:(3)<x>=<f(x)>≈f(<x>)

This simplification procedure is equivalent to the one used in reaction rate theory for chemical reactions [13]. The evolution of cluster size is now given by a deterministic equation in which all fluctuations, i.e., higher moments of P(x,t), are neglected in the function *f*. The Master Equation may then be understood as a mean field approximation.

Assuming the production rate of point defects are uniformly distributed, dynamics of point defects, i.e., vacancies(v) and interstitials (i), can be modeled by the following quasi-equilibrium “chemical” reactions [16]: (4)$$\begin{array}{c} \frac{dC_{i}}{dt}=K_{i}-K_{iv}C_{i}C_{v}-K_{vi}C_{i}C_{s}\\ \frac{dC_{v}}{dt}=K_{v}-K_{iv}C_{i}C_{v}-K_{vi}C_{v}C_{s} \end{array}$$
where “s” refers to the different sinks (free surface, grain boundaries, dislocations loops, cavities) associated with the micro-structure. $K_{i(v)}=\nu_{i(v)}\sigma^{d}\phi$ where ϕ is the flux of impinging particles, $\sigma^{d}$ is a displacement cross section and is an estimation of the displacement cascades formation averaged over all crystalline direction, $\nu_{i(v)}$ an efficiency factor varying from few percents to few tens of percent depending on the temperature and the displacement cascade density. The vacancy-interstitial recombination term $K_{iv}=4\pi r_{iv}^{2}\frac{\left(D_{i}+D_{v}\right)}{\Omega }$ where $r_{iv}$ is the radius capture in which a vacancy and an interstitial recombines and $D_{i(v)}$ is the thermal diffusion of the point defect in the medium and Ω the atomic volume within the effective medium theory framework [9]. The term $K_{i(v)s}$ can also be computed. In the assumption of the equivalent medium [9], it reduces to $k_{i(v)s}^{2}D_{*}(c_{i(v)}-c_{i(v)}^{eq}\left(T\right)$ where $k_{i(v)s}^{-1}$ characterized the mean free path associated with the diffusion of a point defect to a sink and has the dimension of a length and $c^{eq}\left(T\right)$ is the equilibrium concentration of point defect in the medium (for a summary of coefficients used in Equation ([4]), see the article of Bullough et al. in [10]).

Similar equations can be extended to study the growth rate of radiation-induced precipitates or dislocations. This is achieved by adding evolution laws [5] leading to a set of differential equations for the radius of precipitates *R* or the density ρ of dislocations. Within the framework of the rate theory [10,16], the late time evolution of the microstructure is given by the resolution of such a set of differential equations for which the temperature, the flux, the diffusion coefficients or the $k_{i(v)s}^{-1}$ terms, usually called the sink’s strengths, are external parameters, given by different models. In the theory of dynamic systems [17], these parameters ate called as control parameters and they control the possible bifurcation of the dynamical systems to different steady states named attractors. Solving the set of (partial) differential equations defining the dynamical system is often hopeless and need lots of computational resources. Knowing the list of possibles attract, i.e., steady states, allows the prediction of all possible radiation-induced microstructures able to emerge at late time. As the kinetic energy of incident particles (ions, neutrons, electrons) is dissipated into the formation of displacement cascades under irradiation, materials under irradiation belong to the universality class of dissipative systems. The energy dissipation leads to the formation of microstructure, unpredictable from the analysis of the thermodynamic phase diagram. The number and the nature (attractive, repulsive, etc.) of these attractors and the dependence of the long-term behavior of the system to its initial conditions and control parameters accessible to the experimentalists can then be achieved by an inspection of the set of (partial) differential equations [2,18]. For instance, the qualitative evolution of the system can be obtained from the simple computation of the phase portrait as displayed in Figure 1.

The phase portraits of the dynamical system, displayed in Figure 1, for instance, describe the evolution of a small number of variables and are appropriate for systems with small spatial extension (the mean free path $k^{-1}$ is smaller than the distance between sinks). For larger systems, transport processes and/or non local (elastic) interactions can take place:(5)$$\frac{dX}{dt}=G\left(X,\mu \right)+D\nabla^{2}X$$
where *X* is a vector describing the variables of the problem ($c_{i},c_{v}{,}_{r}ho,R$). This equation can be considered as the archetype of Turing reaction-diffusion equations. Introducing in the last term of Equation (Equation 4) the divergence of fluxes $J_{i}=\sum_{j}D_{ij}\nabla X$, such an equation can describe the evolution of the microstructure defined by *X* in large systems with appropriate boundary conditions.

### 2.3. Limitations of the Rate Theory Approach

Despite that rate theory is extensively used to describe the evolution of the microstructure of materials under irradiation, it suffers from two distinct limitations. Since atomic clustering in materials under irradiation is driven by the concentration of point defects (vacancies, interstitials), the complete distribution of clusters (di-mere, tri-mere, etc.) is thus given by a coupled hierarchy of discrete equations. This results in the cluster distribution not only being determined by the average size of clusters <xP(x,t)>, but also by all the different moments of P(x,t). only in a pure birth process, i.e., W(x+δx,x)=cte and other *W* term are null, P(x,t) is given by a Poisson distribution ( $\lambda_{i}=\lambda ;\mu_{i}=0$) and the variance of the process is equal to its mean value. No theorem ensures that in radiation damage, *W* terms follow a Poisson distribution. If it is not the case, fluctuations (second and higher moments of P(x,t)) need to be explicitly taken into account even in the averaged equation. A hierarchy of equations describing the evolution of the different moments of P(x,t) must be computed. In these equations, the moments of P(x,t) are related to those of P(x+1,t). To obtain solutions, we thus need to perform some kind of truncation. Realistic truncation, if it exists, is obviously related to the system size and the correlation length over which two parts of the system “feel” each others. This leads to a drastic increasing of the set of differential equations and obviously the number of control parameters, restricting the predictive power of the rate theory.

Even if the rate theory offers the fascinating possibility to determine different steady states for the microstructure depending values of control parameters, underlying physically sounded mechanisms responsible for the evolution of the microstructure remains unclear. No symmetry argument can be applied to select some of all possible microstructures and up to now model parameters may be force fitted to experimental data. The large amount of fitted parameters for reaction constants limits the predictive nature of such an approach.

Moreover, the continuum theory of elasticity needs to be introduced in the rate theory to discuss diffusion-controlled kinetics due to interactions with applied fields.

## 3. The Phase Field Approach

The main difficulties of the rate theory is due to the fact that many parameters are needed to efficiency model the large time microstructure such as the strength of sinks $k_{*}$ for instance. These parameters evolve with the microstructure via elastic (or electric for ceramics) interactions leading to a feed back effect difficult to understand.

On the other hand, the spatial distribution of sub cascades responsible for the anisotropy of the point defects production terms are difficult to take into account in such a formalism. Kinetic Monte Carlo methods were then developed handling the spatial anisotropy of the defects formation. However, the accuracy of these simulations is directly related to the knowledge of main defects produced under irradiation and their associated diffusion paths. Up to now, no clear formalism such as for instance the detailed balance in thermal equilibrium exists under irradiation, assessing the validity of the choice of the set of defects and their diffusion paths.

During the last decade, the phase field approaches initially developed by Landau [20] for describing phase transition in thermal equilibrium, has been successfully applied for understanding and modeling radiation-induced nano-patterns. This approach seems to reproduce kinetic Monte Carlo simulations and offers a guide to understand key parameters responsible for the nano-structures induced by irradiation. On the other hand, this approach provides the unique opportunity to compute generalized susceptibilities (elastic constants, dielectric constants, etc.) associated with radiation-induced nano-structures. We briefly describe the application of the phase field to radiation-induced patterning.

### 3.1. Foundation of the Phase Field

As it is the case for the rate theory, the phase field may be understood as a mean field theory. To understand this, we can assume that an atom in a disordered solid occupies a vertex of a geometric lattice, assumed cubic for simplicity. At the atomic scale, all sites of the underlying lattice are occupied by species (atoms or vacancy). Calling $\sigma_{i}$ the occupation number at the i -th site and d the dimension of the lattice defined by a characteristic length a, we can define “d-blocks”. Every “d-block” will have a volume $la^{d}$. We can define the averaged occupation σ(r) by a coarse-graining procedure:(6)$$\sigma \left(r\right)=\frac{1}{N_{l}}\sum_{i}\sigma_{i}={\left(\frac{a}{la}\right)}^{d}\sum_{i}\sigma_{i}$$

$N_{l}$ can be understood as the number of degrees of freedom. Obviously, different coarse-graining procedures can be used instead of a simple summation. However, this definition seems reasonable as long as $N_{l}$ is large. After performing an averaging, $\sigma_{l}\left(r\right)$ does not fluctuate much on microscopic scale but varies smoothly in space.

Of course, in general we need to specify *l* in order to determine $\sigma_{l}\left(r\right)$, but the coarse graining procedure we are applying will be useful only if the final results are independent of *l* at least in the spatial scales considered. Under equilibrium, we now must express the partition function in terms of $H_{eff}\left(\sigma_{l}\left(r\right)\right)$, where $H_{eff}$ can be considered as an effective Hamiltonian [21]. Since we now have a system made up of “blocks” this effective Hamiltonian will be composed of two parts: a bulk component relative to the single blocks and an interface component relative to the interactions between the blocks.

When we suppose that every block of volume ${la}^{d}$ is separated from the rest of the system, $\sigma_{l}\left(r\right)$ inside every one of them is uniform only if the dimension of the blocks is much smaller than the correlation length, i.e., the characteristic length of variation of $\sigma_{l}\left(r\right)$ labeled ξ. When la<<ξ, a non equilibrium free energy density f(σ(r)) may be introduced according to Landau [22,23]. This free energy density, f(r), describes the evolution of a uniform system made of non interacting blocks. The coarse graining procedure insures that $\sigma_{l}\left(r\right)$ results from a mean field theory.

To go a step beyond, we now must take into account the fact that adjacent blocks do interact. Since, as we have stated, $\sigma_{l}\left(r\right)$ does not vary much on microscopic scales, the interactions between the blocks must be such that strong variations of magnetization between neighboring blocks are energetically unfavorable. If we call $\delta_{l}$ a vector of magnitude la that points from one block to a neighboring one, the most simple analytic expression that we can guess for such a term can be an harmonic one:(7)$$H_{eff}^{int}=\sum_{r}\sum_{\delta_{l}}\frac{\bar{\kappa }}{2}{\left(\sigma_{l}\left(r\right)-\sigma_{l}\left(r+\delta_{l}\right)\right)}^{2}$$
where the factor 1/2 multiplying $\bar{\kappa }$ has been inserted for convenience. The term $\bar{\kappa }$ defines the stiffness of the material under consideration, i.e., the ability of the material to create interfaces between blocks.

This term known as the Ginzburg term [20,23] can also be considered as a first approximation of a general interaction between the blocks, namely as the first terms of a Taylor expansion of the real interaction energy. Since the linear dimension of the blocks la is much smaller than the characteristic length L of the system, we can treat r as a continuous variable and thus substitute the sum over r with an integral:(8)$$\sum_{r}\to \frac{1}{{\left(la\right)}^{d}}\int dr$$

Finally, we obtain the following expression for the Hamiltonian:(9)$$\begin{array}{c} H_{eff}=\frac{1}{{\left(la\right)}^{d}}\int [f(\sigma_{l}\left(r\right)+\frac{\bar{\kappa }}{2{\left(la\right)}^{d-2}}|\nabla \left(\sigma_{l}\left(r\right){|}^{2}\right]dr^{d} \end{array}$$

Application of the saddle point approximation allows computing the partition function Z(β) and allows identifying $F=-k_{B}Tlog\left(Z\right)$ to the stationary value σ(r) of the functional $H_{eff}$. The term $|\nabla (\sigma_{l}\left(r\right){|}^{2}$ insures that the Ginzburg–Landau free energy *F* may take into account inhomogeneities of the order parameter σ(r)[22,23]. The term $\kappa =\frac{\bar{\kappa }}{2{\left(la\right)}^{d-2}}$ is the cost associated with the formation of an interface. The Landau free energy describes the evolution of the system not only at equilibrium but also far from equilibrium. It must be kept in mind that the non equilibrium Landau free energy exhibits some minima corresponding to those associated with phase diagrams expected from equilibrium. As the free energy is not restricted to the equilibrium free energy, such an approach allows computing a kinetic path for the transition. As it is the case for the rate theory, application of the PF approach to radiation damage may allow computing radiation-induced microstructures.

The kinetics of the system is now given by a partial differential equation:(10)$$\frac{\partial \sigma (x,t)}{\partial t}=\nabla^{p}\int L\left(\sigma \left(x-y,t\right)\right)\nabla^{p}\frac{\delta F(T,\sigma (y,t)}{\delta \sigma (y,t)}$$

L(σ(x,t) is the Onsager coefficient associated with the kinetic path [24]. This kinetic coefficient defines the relaxation time scale. From the definition of the order parameter (atomic position, atomic fraction, polarization, strain field, etc.) and the non equilibrium free energy, more generally thermodynamic potential, it is possible from Equation (Equation 10) to infer some features of the system: list of all possible low symmetry phases for a given symmetry of the “parent” phase, degeneracy of the “low symmetry” phases (number of variants) and susceptibilities (latent heat, heat capacity, etc.). In Equation (Equation 10), $\frac{\delta F(T,\sigma (x,t)}{\delta \sigma (y,t)}$ has a clear meaning. It is the chemical potential of the system. Kinetics occurring during a lower time scale cannot be handle by such relaxation equations. For non conservative OP [22], *p* is null. It is equal to 1 for conservative OP [25]. Neglecting the dependence of Onsager coefficient with the OP, classical PF equations are recovered [26]:(11)$$\begin{array}{c} \frac{\partial \sigma (x,t)}{\partial t}=-L\frac{\delta F(T,\sigma (x,t)}{\delta \sigma (x,t)} \end{array}$$(12)$$\begin{array}{c} \frac{\partial \sigma (x,t)}{\partial t}=M\Delta \frac{\delta F(T,\sigma (x,t)}{\delta \sigma (x,t)} \end{array}$$

The first equation holds for non conservative OP associated with ordering whereas the second equation holds for conservative OP associated with phase separation. *L* can be understood as a damping coefficient controlling the time relaxation of the OP to its equilibrium value. Usually, the term *M* is the mobility and is related to the diffusion coefficient of diffusing species. Introducing the $H^{-1}$ inner product defined by $<f|g{>}_{H^{-1}}=<{\left(-\Delta \right)}^{-1}f|g{>}_{L^{2}}$ for functions of $L^{2}\left(R^{d}\right)$ (d is the dimension of the embedded space), the two equations are formally identical. Both dynamics are then gradient flows of the same Lyapunov functional, L[σ],associated with the $H^{-1}$ or the $L^{2}$ metrics. Computing the inner product in the Sobolev space $H^{-1}$ is equivalent to solve the Poisson equation Δw=−f associated with boundary conditions.

Equation (13) may be derived from the application of the principle of minimal energy dissipation postulated by Onsager [27]. This principle is based on the assumption that the system under consideration is close to an equilibrium condition, so that locally thermodynamic equilibrium can be assumed. As a consequence the concepts of equilibrium thermodynamics can be extended to these non-equilibrium conditions based on the linearization of different fluxes as a function of the generalized thermodynamic forces.

Assuming kinetic coefficients *L* and *M* do not depend on the order parameter, Equation (Equation 13) is a gradient systems and F(T,σ(r,t) is the Lyapunov of the system [13] and $\frac{dF}{dt}<0$. This implies that dynamics evolves to different minima of F(T,σ(r,t). As it is the case for the rate theory, different minima of F(T,σ(r,t) define a set of different attractors of the system. These attractors are separated by separatrices. These separatrices themselves make up an invariant set for the gradient flow dynamics. These gradient systems are dissipative systems since $\frac{dF}{dt}<0$ even when *M* and *L* are assumed independent of the order parameter.

These partial differential equations can be solve as soon as boundary and initial conditions are defined:The initial condition is usually given by an uniform distribution $\bar{\sigma }$ of the order parameter in the integration volume *V*. Since $\bar{\sigma }$ is an attractor of these equations, a small random variable db is added to the uniform value of the order parameter. The mean value of this random variable db is null and its variance of the order of $k_{B}T$ [28].For close systems, Neumann conditions must be applied to the the order parameter and the driving force, defined by the chemical potential $\frac{\partial L[\sigma ]}{\partial \sigma (x,t)}$. This implies ∇σ ad $\nabla^{3}\sigma$ are null over δV for the conservative Cahn–Hilliard equation. Obviously for the non conservative equation, only the second condition holds.Neglecting edge effects, periodic conditions are usually applied for computing microstructures within the framework of the phase field theory. This implies that the order parameter σ(x,t) can be expanded in an infinite cosine set. Numerically, a finite set of discrete cosine transform (dct-I) is used to solve these equations for close systems [29].

### 3.2. Application to Radiation Damage

Applying the PF approach for computing radiation-induced pattern implies to take into account the effect of radiation damage. Radiation damage is handled in two distinct ways:Point defects produced during the collision step are assumed to only modify the mobility of species or more precisely the kinetic Onsager coefficients. Their concentration is supposed to evolve rapidly with time and reach a steady state. It may result from the resolution of RT equations. The interest of such an approach lies in the fact that this steady state may evolve with the nano-structure dictated par the solutions of the Phase Field approach. The nano-structure is no more freeze and defined at the beginning of the simulation such as in the RT but can evolve with time. Under this approach, the atomic fraction of point defects is always small and does not modify the value of the thermodynamic potential describing the formation of the nano-structure under equilibrium. For instance, the value of the stiffness coefficient trigging the formation of heterogeneous structures does not depend on the atomic fraction of point defects.The motion of atoms under irradiation is dictated by an external dynamics resulting only from the ballistic mixing occurring inside sub-cascades. This ballistic mixing is due to a spatial averaging of atomic collisions occurring inside the sub-cascade. The flux of ejected particles is not proportional to the chemical potential of different species in the material as it is the case in thermal diffusion but depends on the atomic fraction of diffusing species. The key feature of the ballistic mixing lies on the fact that it exhibits a spatial dependence. As clearly pointed out by Sigmund [11], it depends on the nature and the energy of incident particles and can be modeled by a simple diffusion equation. It can be analytically computed [1] or extracted from MD simulations [30].

As a first approximation, the dynamics induced by the ballistic mixing may be modeled by the following equation:(13)$$\frac{\partial \sigma }{\partial t}=\Gamma \left(E,\phi \right)\int \left(p_{R}\left(y\right)-\delta \left(y\right)\right)\sigma \left(x-y\right)dy$$

The term Γ(E,ϕ) defines the strength of the the mixing, i.e., the ability of irradiation to exchange atoms at a given position irrespective to their chemical nature. It is a function of the energy of primary knock-on atoms *E* and the flux ϕ of incident particles. $p_{R}x)$ is a probability density function modeling the ability to eject atoms at a distance x from its initial position in the sub-cascade. Generally speaking, this probability density function may be modeled by a Levy’s function [31]. The term *R* can be understood as the mean free path associated to the ballistic mixing. It is a complex function of the energy and the mass of incident particles and atoms forming the material. For incident particles with kinetic energies below a few MeV, *R* is of the same order of magnitude than the unit cell parameter [30].

Application of the phase field theory to radiation-induced microstructure implies that *R* must be larger than the coarse graining characteristic length la. Fixing the coarse-graining length to *R* implies that the PF approach is not able discussing modifications of interfaces below *R*. The PF approach then only provides a description of the microstructure over length scale larger than few nanometers forbidding the description of radiation-induced dislocations loops for instance.

Within the framework of the phase field theory, the radiation-induced microstructure results from a competition between thermal and ballistic effects. The first one is associated with the thermal evolution of the system and modeled by a PF equation and the second one results from a ballistic mixing generated inside sub-cascades:(14)$$\begin{array}{ccc} \frac{\partial \sigma (x,t)}{\partial t} & = & M\left(T,\phi \right)\Delta \frac{\partial F(\sigma (x,t))}{\partial \sigma (x,t)}\\ & + & \Gamma \left(E,\phi \right)\int (p_{R}\left(y\right)-\delta \left(y\right))\sigma \left(x-y\right)dy \end{array}$$
where σ(x,t) is the order parameter proportional to the local atomic fraction of the species *B* in the AB alloys. This order parameter is by definition null at high temperature. The mobility M(T,ϕ) enhanced by the steady state concentration of point defects produced under irradiation appears to be the unique control parameter. Equation (Equation 14) illustrates the power of the phase field approach. Even if the late time evolution is modeled by a partial differential equation, as it is the case in the rate theory, only a unique control parameter physically sounded M(T,ϕ) is needed.

Moreover, Equation (Equation 14) allows computing a Lyapunov functional for the system $L\left(t\right)=F\left[\sigma \left(x,t\right)\right]+\frac{\Gamma (E,\phi )}{2M(\sigma (x,t)}G\left[\sigma \left(x,t\right)\right]$ where $G\left(t\right)=\frac{1}{2}\int \sigma \left(y,t\right)g_{R}\left(x-y\right)\sigma \left(x,t\right)dxdy$ and $\Delta g_{R}=p_{R}\left(\sigma \left(x\right)\right)-\delta \left(x\right)$. The function $g_{R}\left(x\right)$ quantifies the “averaged” spatial extension of the ballistic mixing induced by all sub cascades in the medium. As $p_{R}\left(x\right)$ displays an exponential-like tail associated with rare events, i.e., the ejection of atoms far from the center of the sub cascade [31], $g_{R}\left(x\right)$ is also a decreasing function of x as expected. Assuming the ballistic mixing occurring in sub-cascades is isotropic and $p_{R}\left(x\right)=p_{R}\left(x\right)=\propto exp\left(-\frac{x}{R}\right)$, $g_{R}\left(x\right)$ is isotropic and can be explicitly computed. A similar Lyapunov functional can be computed for non conservative OP [32].

The key point of this analysis is that $\frac{dL[\sigma (r,t)]}{dt}<0$. This inequality insures that the system is dissipative as expected. The energy of incident energetic particles is dissipated in many collisions during the slowing down of these particles in the medium and generate nano-patterns. The existence of such a Lyapunov implies that no chaotic nor oscillatory steady state exists [33] and its minimization allows drawing all possible steady states produced under irradiation. This Lyapunov functional L[σ] can be understood as an “effective” free energy including irradiation damage. However, recent numerical simulations [34] highlight the fact that radiation-induced patterns in non miscible binary alloys depend on the average composition of the alloy as displayed on Figure 2.

This result, in agreement with kinetic Monte Carlo simulations [35], highlights fundamental differences between systems under equilibrium and far from equilibrium. Such a dependence of radiation-induced steady states with the average composition forbids to describe the evolution of the microstructure in irradiated alloys applying the notion of phase diagram [4,35,36,37], as it is the case under thermal equilibrium. The existence of an “effective” free energy is not sufficient to apply concepts of the thermal equilibrium to systems far from equilibrium. In fact, this functional is not homogeneous under irradiation but varies in space. Ground states of this functional are not homogeneous forbidding to establish any phase diagram under irradiation.

This example shows that the phase field offers the possibility to list all possible radiation-induced steady states tracking minima of the Lyapunov. A Lyapunov functional may also be computed from the rate theory formalism even if its computation is much more complicated and many parameters need to be known. Both theories then give the same information on the final microstructure. The main advantage of the phase field approach is its ability to propose a mechanism for the formation of radiation-induced patterns.

The self-organization of systems leading to the formation of patterns results from the invariance of the correlation function versus the space scale [13,26,33]. Computing the structure factor $S\left(k,t\right)=<|\hat{\sigma }{\left(k,t\right)|}^{2}>$ (where <.> is a radial average and $\hat{f}$ is the spatial Fourier transform of f(r)) is the usual way to track the spatial invariance. Figure 3 illustrates this point showing structures factors (right panel) associated with a random (top graphs) and a self-organized (bottom graphs) distribution of Cu-rich precipitates in AgCu [38]. Within the phase field theory, $S\left(k,t\right)\propto {\left(\frac{\partial L^{\prime }\left[\sigma \right]}{\partial [\sigma ]}\right)}^{-1}$ is directly related with the quadratic part of the Lyapunov via $L^{\prime }\left[\sigma \right]=\int D\left(k\right){\left|\hat{\sigma }\left(k,t\right)\right|}^{2}dk$ with $D\left(k\right)=[-1+k^{2}+\Delta \left(T,\phi \right)g_{R}\left(k\right)]$.

The term Δ(T,ϕ) is the ratio between the a-thermal ballistic coefficient Γ(E,ϕ) and the radiation-enhanced mobility M(ϕ,T) and appears to be the unique control parameter of the problem:Low values of Δ(T,ϕ) implies that ballistic effects induced by irradiation can be neglected and that the system behaves as it does under thermal equilibrium. S(k,t) displays a maximum at $k_{0}\propto t^{-1/3}$ as expected for a spinodal decomposition [24].High values of Δ(T,ϕ) implies that ballistic effects dominate the micro-structure of the material under irradiation. As thermal effects are negligible, the large time microstructure results from a random distribution of Cu atoms in an Ag matrix. S(k,t) displays a maximum at $k_{0}=0$.When Δ(T,ϕ)≈1, a competition between thermal and ballistic effects occurs. At large time, S(k,t) displays a maximum at a non null $k_{0}$ value independent of time [39]. The pattern shows the existence of this non null $k_{0}$ value of about few ${\mathrm{nm}}^{-1}$. This implies that a nano-patterning occurs under irradiation for given values of temperature *T* and flux ϕ.

A large time, D(k) has a minimum $k_{0}$ and can be expanded in the neighborhood of this minimum:(15)$$\begin{array}{ccc} D(k) & = & D\left(k_{0}\right)+\frac{d^{2}D\left(k_{0}\right)}{2dk^{2}}{\left(k-k_{0}\right)}^{2}\\ & \approx & D\left(k_{0}\right)+\frac{d^{2}D\left(k_{0}\right)}{8k_{0}^{2}dk^{2}}{\left(k^{2}-k_{0}^{2}\right)}^{2} \end{array}$$

Since $k_{0}$ is a minimum of D(k), $\frac{d^{2}D\left(k_{0}\right)}{dk^{2}}$ is positive and the first order expansion of D(k) leads to the conservative form of the Swift-Hohenberg equation [33]. This equation is extensively applied to discuss liquid/crystal transition in unary systems [40] It was successfully used to model dynamics of microstructural defects [41], diffusive atomistic dynamics of edge dislocations [42,43], plasticity and micro-mechanics of emergent patterns in plastic flows [44] out of irradiation.

This example illustrates how the Phase Field can help to rationalize the appearance of radiation-induced nano-patterns in irradiated materials. Therefore, more information can be deduced from the phase field model. It may explain the formation of nano-patterns. In the example of the AgCu alloy, the minimum of D(k), occurring for Δ(T,ϕ)≈1, is isotropic. However, ground states of L[σ(r)] describing radiation-induced microstructures are non isotropic symmetry adapted functions for satisfying boundary conditions. The mechanism associated with the formation of nano-patterns can be clarify. It results from the symmetry breaking of L[σ(r)] by ground states. This symmetry breaking generates an instability such as the Eckaus instability [5]. This instability propagates via Goldstone modes [45], i.e., without dissipation of energy [2].

The existence of nano-structures displayed in Figure 2 generalizes the notion of topological defects defined for equilibrium structures and are generally the rule beyond pattern forming instabilities, as it is the case in equilibrium phase transitions [26,33]. In graph c of Figure 2, disclination and geometrical dislocation are clearly visible.

The phase field approach points out that defects result from the existence of a set of unstable modes. Although each individual spatio-temporal patterns break transnational and rotational symmetries, different patterns grow in different part of the system implying the formation of various defects connecting these different steady states.

The formation of energy-less nano-structures via the propagation of Goldstone modes impacts the properties of materials. In a more precise way, the appearance of radiation-induced nano-structures may modify the response of the materials to external perturbation such as constraints or temperature gradients. By its ability to compute generalized susceptibilities [23], the phase field method then offers the unique opportunity to compute for instance variations of elastic constant induced by the formation of B-rich precipitates under irradiation [46]. As irradiation stops the coarsening of B-rich precipitates in non miscible alloys [39], the characteristic size of nano-structured B-rich precipitates is about few nanometers in agreement with experimental results [46,47] in AgCu and are always in coherence with the underlying atomic lattice. Applying the formalism developed by Katchaturyan [48] for computing elastic constants, variation of elastic constants resulting from the radiation-induced microstructure has been computed. Figure 4 displays the comparison between Young moduli computed from the application of the Katchaturyan’s methods [46] and extracted from nano-indentation [49] experiments performed in $Ag_{0.42}Cu_{0.58}$ thin films irradiated at different temperatures (for all experiments, the Poisson ratio is equal to 0.24 in agreement with numerical simulations). The experimental increasing of about 10% of the Young modulus with the temperature (navy triangle on Figure 4) can only be modeled taking into account the precipitate-precipitate dipoles elastic interactions (blue stars on Figure 4) highlighting the impact of infinite range on the elastic properties of irradiated alloys. Neglecting this interaction leads to a decreasing of the Young modulus with the temperature (red line on Figure 4) not experimentally observed.

### 3.3. Limitations of the Phase Field Approach

Despite that the phase field method seems to reproduce nano patterns observed in some irradiated alloys, its application to radiation damage is still questionable:The theoretical foundations of phase field models lies in the ability to perform a coarse-graining procedure over a length scale la small in comparison with the characteristic length of the system *L* but large enough to neglect fluctuations of the order parameter proportional to ${\left(la\right)}^{d/2}$ under equilibrium. Under irradiation, specific mechanisms of energy deposition may impose larger fluctuations over this length scale and then may be taken into account to define the coarse graining length scale. Whatever the choice of the coarse graining length, this approach is inaccurate to compute for instance the formation of radiation-induced dislocation nor evolution of the short range order.The control parameter Δ(ϕ,T) is the ratio between a-thermal ballistic diffusion term and the radiation-enhanced mobility M(ϕ,T). This last term is directly related with the concentration of point defects produced under irradiation. This concentration depends on the late time microstructure, resulting from the calculation of the microstructure using the phase field approach, via the numbers and the density of sinks. This feedback effect must be studied in detail.Since the dynamics of vacancies and interstitials created under irradiation cannot be described by any free energy functional, the phase field approach is inefficient for computing the evolution of point defects under irradiation.

Many authors apply the phase field approach for modeling the evolution of point defects under irradiation [50,51]. They combine a Landau–Ginzburg functional to describe the spatial evolution of the concentration of point defects and a rate theory approach. However, the use of a Landau–Ginzburg term for describing agglomeration of vacancies is doubtful. Recent publications [52] confirm the analogy between the Swift–Hohenberg equation extensively used for describing out of equilibrium structures and the PF approach of ballistic mixing modelled by Equation (Equation 14) firstly pointed by Simeone et al. [38]. However, they pointed out that this analogy may not be satisfied in all the patterning regime. Recent works [34] confirm that patterns similar to those derived from the SH model can be observed in the pattern domain and that approach extends also to the phase separation regime induced by low irradiation fluxes.

## 4. Conclusions and Perspective

Studies of the behavior of materials maintained far from equilibrium at length scale larger that what we can handle by direct atomic simulations and smaller than what allows macroscopic continuum averaging displays difficulties. The fair agreement between simulation and experimental results demonstrates that the PF approaches is then a powerful tool able to simulate not only microstructure but also properties of irradiated materials. Application of this method allows predicting the unexpected patterns observed in non miscible alloys such as AgCu [47] under irradiation at large time. This method demonstrated that the notion of phase diagram cannot be applied under irradiation. By its ability to compute generalized susceptibilities, it also offers the unique opportunity to understand and explain modifications of properties of materials under irradiation.

The main interest of this mean field approach in comparison with the rate theory formalism, extensively used to describe the evolution of the microstructure in irradiated materials, lies in the small physically sounded number of control parameters in comparison with those used in the rate theory. The strength of this approach is also its weakness. Since the phase field approach is based on a coarse graining procedure and a minimization of an “effective” free energy density, it fails to model evolution of radiation induced point defects.

The existence of kinetic Onsager’s coefficients, i.e., *L* and *M*, under irradiation as well as their dependence with the defect concentrations need to be clarified. In particular their dependence to the the order parameter will dictate the dynamics of the system. Since these coefficients can be considered as independent of the order parameter, the system is is gradient flow system and at equilibrium.

On the other hand, both rate theory and Phase field approach assume that reaction paths (meta-sable states and minima of the energy landscape) are independent of the environmental noise which does not reduced to $k_{B}T$ under irradiation. This point needs to be understood in detail [53].

The combination of rate theory and phase field should give an accurate description of the microstructure of material under irradiation. To be achieved, efficient methods need to be developed to plug these two approaches. Combination of rate theory and PF could achieve both the computation of a realistic population of point defects and nano-patterns at large time scale. Extension of this method to irradiation by swift heavy ions which mimic the impact of fission products and to oxidation processes under irradiation, is still an open field.

## Figures and Tables

**Figure 1 materials-15-02991-f001:**
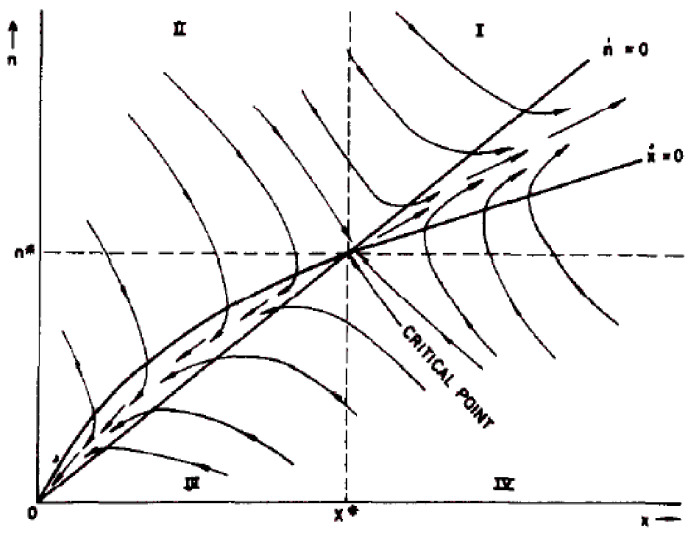
Example of phase portrait of solute atoms in precipitates (x) and vacancies (n) calculated under irradiation. Reprinted with permission from Ref. [19]. Copyright 1977 Elsevier. At large time, the system can reach two distinct attractors: a stable node (0,0) and a the saddle critical point (x*,n*). The existence of these attractor partition the abstract x-n plane in four distinct quadrants. The arrows displays the velocities of the solute atoms and vacancies in each quadrant (I to IV).

**Figure 2 materials-15-02991-f002:**
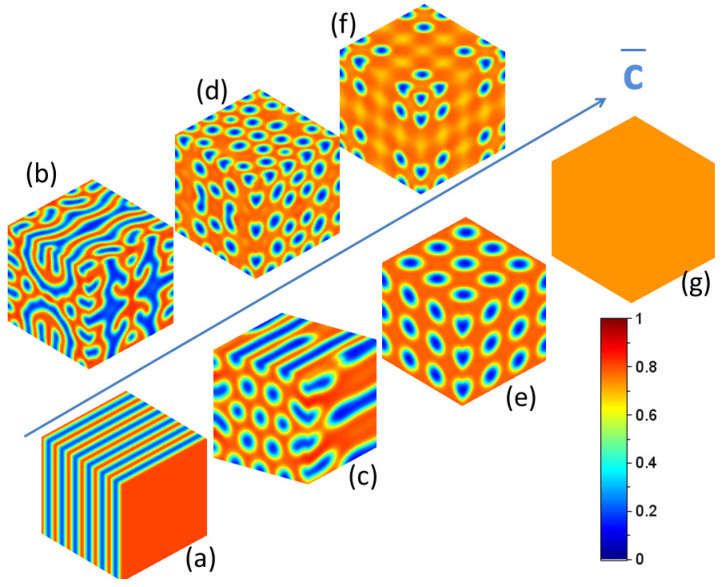
Simulation of 3D nano-patterns of $Ag_{\bar{c}Cu_{1-\bar{c}}}$ irradiated with 1 MeV Kr ions at 330 K with a flux of $\phi =10^{11}{\mathrm{cm}}^{-2}s^{-1}$. Different periodic microstructures emerge depending upon the average Ag concentration $\bar{c}$ ((**a**): 1D stripes (S); (**c**): cylinders forming hexagonal patterns (C); (**e**): 3D body centered cubic structures ($\bar{c}=0.65$) (B); (**g**) solid solution ($\bar{c}=0.75$) (SS)). Micro-structures associated with the co-existence of such domains are also plotted ((**b**): S+C; (**d**): C+B; (**f**): B+SS). The color stick defines the composition of radiation-induced nano-patterns. The blue and red colors refer to a null concentration of Ag and a null concentration of Cu respectively.

**Figure 3 materials-15-02991-f003:**
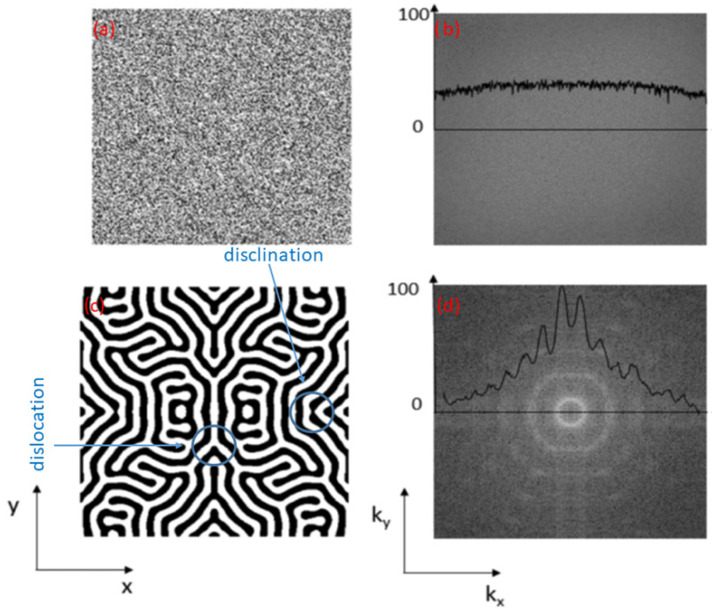
Radiation-induced patterns simulated via the phase field approach of $Ag_{0.38}Cu_{0.62}$. The initial solid solution (graph **a**) leads to a labyrinthine structure (graph **c**) after irradiation with 1 MeV Kr ions at $\phi =6\times 10^{12}{\mathrm{cm}}^{-2}s^{-1}$, and T=440 K. Associated Power Spectral Density ${\left|S\left(k,t\right)\right|}^{2}$ in the Fourier space are also plotted (graphs **b**,**d**). In these simulations, A rich (black) (B rich (white)) domains are associated with positive (negative) σ values (the 3D domain size is equal to 200 in reduced units). The PSD is radially symmetric and peaks sharply around $k_{0}$ (intense white ring in graph **d**) as expected for a patterning whereas it is flat as expected for a random distribution of atoms (graph **b**). On the 2D cut of the 3D pattern (graph **c**), extended defects such as disclination and geometrical dislocation lines are clearly visible.

**Figure 4 materials-15-02991-f004:**
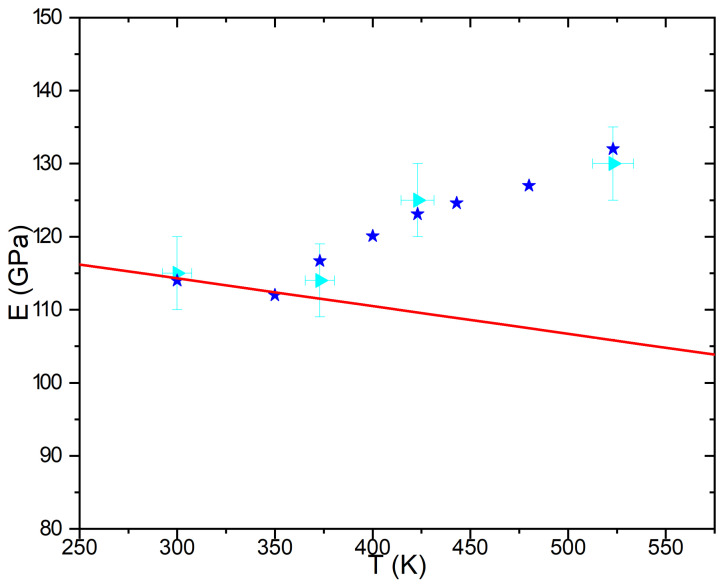
Comparison between experimental Young moduli (navy triangles) extracted from nano-indentation experiments performed on $Ag_{0.42}Cu_{0.58}$ thin films and computed from application of the Phase Field approach including elastic terms (blue stars) versus the temperature. The red line is displays Young moduli resulting from a simple lever rule neglecting the formation of nano-structures under irradiation.

## Data Availability

Not applicable.

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
