# Peer review of "Radiation-Induced Patterning at the Nanometric Scale: A Phase Field Approach"

_materials, 2022, doi:10.3390/ma15092991_

Round 1
Reviewer 1 Report
materials-1615566
In this review, the authors discuss about Radiation-induced patterning at the nanometric scale: a Phase Field approach. and the resulys are very improtant some Combination of rate theory and phase field.
Combination of rate theory and PF.
Extension of this method to irradiation by swift heavy ions which mimic the impact of fission products and to oxidation processes under irradiation, is still an open field.
The manuscript is written very well. The presented information’s are based on references that are updated; the work is well-structured, well-written and easy to understand. It also addresses a subject that is of great interest in the scientific community. The topic is perfectly in line with the "Materials journal" and therefore I recommend publishing after minor revisions,
Plagiasm = 25% you can rectified more
- However, irradiation with pure ions generates fission products which can be neutrons, protons, trition of pions, why not irradiate with deteron which generates less fission product than ions.
- In your opinion how reliable these approaches are and does not give irradiation donger?
- In Figure 2 you can add more detail about the color sections to understand the intensities or scales.
- Figure 3 you can name them as Fig 3.a, Fig 3.b .... to understand the interpretations.
Reviewer 2 Report
Please see attachment.

Round 2
Reviewer 2 Report
I recommend the paper for publication. A few comments:
- I would refer the authors to take note of the following references where phase field crystal model has been shown to successfully model dynamics of microstructural defects (not for the case of irradiation though).
-
- Diffusive atomistic dynamics of edge dislocations in two dimensions.
- Plasticity and Dislocation Dynamics in a Phase Field Crystal Model.
- Micromechanics of emergent patterns in plastic flows.
- The recently submitted (revised) pdf has the same figures for Figs 2,3,4. Please ensure that the figures are correctly replaced.
- Also, in Line 403, please correct the sentence "It must noticed ....." .
- Also, please correct the spelling of "Lyapounov" to "Lyapunov" in the manuscript.
